# Dynamic Correlation between the Chinese and the US Financial Markets: From Global Financial Crisis to COVID-19 Pandemic

Jianxu Liu [1] , Yang Wan [1,*] , Songze Qu [2,*], Ruihan Qing [1] and Songsak Sriboonchitta [3]

1    School of Economics, Shandong University of Finance and Economics, Jinan 250000, China
2    School of International and Public Affairs, Columbia University, New York, NY 10027, USA
3    Faculty of Economics, Chiang Mai University, Chiang Mai 50200, Thailand
*    Correspondence: 20190611025@mail.sdufe.edu.cn (Y.W.); sq2229@columbia.edu (S.Q.)

**Abstract:** As China's economy and the U.S. economy have shown a definite interaction, there is considerable interest in studying the correlation between the Chinese stock market and the US financial markets. This paper uses an Asymmetric Dynamic Conditional Correlation (ADCC)-GARCH to investigate the correlation between the Shanghai Composite Index (SHCI) and the U.S. financial markets, including SP500, NASDAQ, and US dollar indexes. The empirical results show that the time-varying daily and the lag-one correlation between China and the US stock markets have different performances during global events and national events. Compared with the complicated effect of negative events on the correlation of the stock market, SHCI and USD are negatively correlated with higher negative correlation during the global negative events. In addition, we found Chinese investors are more contagious to the news than American investors, indicating that the Chinese government's policy are more indicated to Chinese investors. Finally, some policy suggestions are provided, and are beneficial to risk prevention and control, and investment.

**Keywords:** dependence; SP500; USD; financial crisis; COVID-19

## 1. Introduction

In line with globalization and technological advances, China's economy and the US economy are now undoubtedly more connected. This close economic partnership has the potential to strengthen the interrelationship between Chinese and U.S. financial markets [1]. In view of the "too connected to fail" theory, research on the correlation of Chinese and American stock markets is of great significance, because if a risk occurs in one of these financial markets, both sides will be hurt. On the other hand, the two largest economic systems are in the US and China, which constitute approximately 42.10% of the world GDP [2]. The US financial market and the Chinese stock market are also firmly in the leading position in the world. The economies and financial markets of China and the US are huge. As the "too big to fail" theory explains, a study of the relationship between the two countries' financial markets is necessary. The stock market as one of the irreplaceable supporting lines for the development of the real economy is then, naturally, a major focus for researchers and investors to actively investigate and put into practice in their portfolio management. The US stock market demonstrates the ability of information flow to global stock exchanges [3]. The U.S. dollar is the world's dominant currency and a major component of U.S. financial markets. The fluctuations of the U.S. dollar have a significant impact on exchange markets and economies in the world. For example, appreciation of the U.S. dollar causes the depreciation of Chinese Renminbi (RMB). This depreciation benefits exporters and could fuel export-driven growth in China, which could lead to the Chinese stock market rally. However, according to the capital flow theory, RMB depreciation will promote international capital outflow and drive stock prices down [4]. Therefore, the U.S. dollar and the Chinese stock market do not have a clear relation. Most importantly, the U.S. dollar index has maintained an upward trend during the COVID-19 epidemic,

which is similar to the financial crisis in 2008. Clarifying the association between the U.S. dollar and the Chinese stock market is potentially beneficial for investors and financial institutions to implement into their portfolio management in a timely and informative manner. Obviously, the analysis of interdependence of the US and China financial market is important for effective diversification in portfolio management. Evidence shows that the co-movement between the global stock market has increased during the China-US trade war [5]. Under the background of the trade war and the global pandemic, the growth of US-China co-movement is slow compared to the coalition of trading partners, but it does show a clearly strengthened connection between the US financial market and China's stock market, more specifically, Shanghai Composite Index (SHCI) and the U.S. financial markets including SP500, NASDAQ and US dollar indexes. To sum up, the analysis of the co-movements between Chinese and American financial markets is of great research significance for investment portfolio and systemic financial risk.

While the investor structures in China's stock market and the US stock market are different, with China's majority investors being individuals with small-mid assets and the US's major participants being financial institutions, the connection between these two markets is unexpected high. This pattern is observed in previous studies which analyzed the pre-US-China trade war period from 2000 to 2010 [6]. In this paper, we evaluated the correlation between the Shanghai Composite Index and the U.S. financial markets including SP500, NASDAQ, and US dollar indexes using a Dynamic Conditional Correlation (DCC) or an Asymmetric DCC-GARCH model. Engle [7] proposed the DCC-GARCH model to measure dynamic correlation between two financial assets, and Cappiello et al. [8] extended it to the Asymmetric-DCC (ADCC)-GARCH model to capture possible asymmetric relationships between two assets. Because these two models are widely used in the financial field (see [9–11]), we used them to measure the dynamic correlation between the Chinese and American financial markets, and adopted the information criterion, e.g., Akaike Information Criterion (AIC), Bayesian Information Criterion (BIC) and Hannan-Quinn information criterion (HQIC), to select the best fitting model. Some evidence show that the US financial market has a significant impact on the global market, including China's stock market, until the trade war [12]. This paper will re-examine the claim and further shed light on the effect during the worldwide pandemic using the ADCC-GARCH model.

Consequently, this paper aims to measure the co-movements between Chinese and the US stock markets, and Chinese stock market and the US dollar from the SARS to the COVID-19 pandemic. This objective not only helps the Chinese government and the US government prevent risk contagion from each other, but also has reference value for financial institutions and investors to make rational portfolios. We use empirical methods to answer the following questions: What is the dynamic correlation between the Chinese stock market and the US financial markets from pre-crisis to the COVID-19 period? Why is there sometimes a low correlation between the two markets? The testing approach for investigating the dynamic correlation between the two markets is to use target daily returns and indexes in the two markets, namely the SHCI, SP500, NASDAQ and USD indexes over 2000–2021. We use the data to estimate ADCC-GARCH models which test the dynamic correlation with controls for other essential factors that affect the returns.

Our key findings are as follows. There is a convincing result showing that the effect of major events on the co-movement between China's stock market and the US stock market depends on the type of the major events depending: (1) during negative national events, daily correlation and lag-one correlation between the two markets usually move in the opposite direction, indicating a unidirectional information transmission; (2) during negative global events, both of the correlations between the two markets decreases and the two markets show independency. These findings suggest that investors should be cautious about transitional hedging during negative global events and that the market regulators and operators should consider externality when making corresponding policies under a major negative event.

This paper is organized with the following structure: Section 2 depicts the associated literature and explains the position of this paper within the pool. Section 3 presents the data and the methodology. Section 4 shows our results and findings, and Section 5 discusses the limitation and portfolio management implications of these findings.

## 2. Literature Review

A large number of studies have indicated that global financial markets demonstrate co-movements due to various causes, and the relationships are becoming tighter in recent years [13–15]. Among the research investigating the correlation and co-movements between stock markets under different jurisdictions, Agmon [16] was one of the earliest studies examining the interaction between the equity markets in different countries using regression analysis. Their findings suggest despite the market segmentation, evidence from the United States, United Kingdom, Germany, and Japan shows that there exist significant correlations between the markets that are studied. A large body of literature studying the co-movements between multinational equity markets uses econometric approaches. It can be seen that the research on relationship of stock markets can be traced back to the 1970s at least, and their research achievement is also rich. Therefore, we divided the Literature Review into three subsections, as follows. Section 2.1 reviews the research on the relationship between the US stock market and other financial assets. Section 2.2 summarizes research on the linkage between the U.S. dollar and other financial assets. Section 2.3 provides an overview of the research on the relationship between the Chinese stock market and other financial markets.

### 2.1. The US Stock Market and Other Financial Markets

The US is the largest stock and financial market, which naturally attracts the attention of a large number of scholars. There is a very fruitful body of research on the relationship of financial markets between the United States and other countries, or other economic indicators. Becker et al. [17] verified that there exists a strong correlation between the US and Japanese stock markets. Cha and Seeking [18] used the vector autoregression (VAR) model to study the association between the US and the Asian emerging stock markets. Utilizing ARCH and GARCH model, Forbes & Rigobon [19] found strong interdependence among stock markets in North America, South America, Europe, and Asia during the 1997 Asian crisis, 1994 Mexican devaluation, and 1987 U.S. market crash. Lahrech and Sylwester [20] empirically measured the U.S. and Latin American stock market linkages by using dynamic conditional correlation (DCC) approach. Sakurai and Kurosaki [21] found that the correlation between the oil market and the US stock market became higher after the COVID-19 pandemic. Chin and Paphakin [22] studied the daily relationship between the U.S. asset prices and stock indices of four American countries. Some studies focused on other kind of relationships between the US stock market and other stock markets or financial assets, such as causality relationship [23,24], and transmission relationship [22,25,26].

### 2.2. The US Dollar and Other Financial Assets

In the global financial market, the U.S. dollar also has an important function. Some scholars and researchers also pay attention to the relationship between the U.S. dollar and other financial assets, such as gold [27–29], crude oil [30], cryptocurrencies [31], and stock markets [32,33]. Most of studies verified that the U.S. dollar has significant movement with stock markets [34,35]. During the COVID-19 pandemic, many scholars paid attention to the influence and changes of the U.S. dollar. Cheema et al. [36] found that the safe-haven character of the U.S. dollar has been weakened compared to the 2008 financial crisis. Wang and Wang [37] demonstrated that the market efficiency of the U.S. dollar sharply decreases during the extreme event of the COVID-19 pandemic. Tran and Nguyen [29] showed that the stock market in Europe had a negative impact on the U.S. dollar in 2021. Liu and Li [38] showed that there exists volatility spillover effect and dynamic correlation between the U.S. dollar and bitcoin, which is magnified with the advent of COVID-19. In general, Black Swan events have an impact on stock markets and the U.S. dollar index. The correlation

between the stock market and the US dollar may be strengthened during a Black Swan event, such as the financial crisis and the COVID-19 epidemic.

### 2.3. The Chinese Stock Market and Other Financial Markets

With the continuous integration of China into the global economy, the Chinese stock market has also received a lot of attention, especially in the last decade [39–42]. Among many studies, relationship between China and the US financial markets is an important research issue. Most of the studies found that there is no, or a weak association between these two stock markets before the 2008 financial crisis (see [43–45]). The reasons are that China's stock market is considered in isolation from the global financial market, and there is no overlap in their trading hours [46,47]. However, various studies' evidence showed that the two stock markets have co-movement at a certain level, and affected each other after 2006 [48,49]. With the outbreak of COVID-19 pandemic, there are few studies focused on the co-movement of the two stock markets during the COVID-19 pandemic. Song et al. [50] found that the Ganger Causality between the stock markets during crisis was significantly higher than the pre-crisis period, while Ben Amar et al. [51] verified that the Chinese stock market does not impact on the US stock market during the COVID-19 pandemic.

By reviewing the above literature, we can identify four important findings. First, the co-movement between the Chinese and US stock markets is not invariant. Second, there are few co-movement analysis results of the Chinese and American stock markets during the COVID-19 epidemic, and the conclusions have obvious differences. Third, due to the impact of the trading hours of the two markets, the correlation of daily return at time $t$ may not be high or representative. Fourth, we affirmed that the Chinese stock market and the U.S. dollar are related, and there have been no studies on the relationship during the COVID-19 pandemic. Therefore, this paper puts forward the following hypotheses: (1) The correlation between the Chinese stock market and the US financial market are time-varying. (2) During Black Swan Events, such as the 2008 financial crisis and the COVID-19 pandemic, the Chinese stock market has an increasing association with the US stock markets. (3) Lag–one correlation between the Chinese stock market and the US financial market is higher than the current one. (4) The correlation between the Chinese stock market and the US dollar are time-varying, and the correlation during the financial crisis and the COVID-19 pandemic is stronger than usual.

This paper contributes to the literature on the co-movements between China's and the US equity markets in the following ways. First, this paper utilized daily return data on the two markets and extended the analysis to the COVID-19 period. This research investigation period from 2000 to 2021, including the global pandemic period, allows us to prepare for the next global plague more comprehensively, from a financial perspective. Second, to the best of our knowledge, this paper innovatively assessed the effect of the SARS and COVID-19 pandemic on both the daily and the lag-one correlations between the two markets and provided empirical evidence on the two markets. Compared with previous studies with a similar research focus, such as [50,52], this paper measures the time-varying correlations between Chinese and the US financial markets during the COVID-19 pandemic, but not a static or volatility analysis.

### 3. Methodology

In order to evaluate the volatility of SP500, NASDAQ, USD, this paper used GJR (Glosten, Jagannathan, and Runkle)-GARCH derived from the GARCH model to calculate the Volatility. Compared to the GARCH model, GJR-GARCH includes a new variable to describe an asymmetry effect of the return of the stock [53]. The variance equation in a simplified GJR-GARCH model can be expressed as follows:

$$h_t = \alpha_0 + (\alpha_1 + \gamma_1 I_{t-1})\varepsilon_{t-1}^2 + \alpha_2 h_{t-1} \tag{1}$$

$$\varepsilon_t = \sqrt{h_t}\eta_t \tag{2}$$

where $\gamma_1$ is the coefficient of GJR which describes the effect of the leverage effect; $I_{t-1}$ is a dummy variable (if $\varepsilon_{t-1} < 0$, $I = 1$, otherwise $I = 0$). $h_t$ standards for conditional variance at time $t$. $\eta_t$ is standard normal distributed or student-t distributed with degree of freedom $\nu$.

We use the DCC-GARCH model to evaluate the dynamic correlation between the Chinese stock market and the US financial markets. By using this model, we can easily and directly account for heteroscedasticity by estimating the correlation coefficients of the standardized residuals [54]. The characteristics of the correlation between return and information spillover effect of the four indexes over time are then captured [55].

Based on Engle [7], the multivariate DCC-GARCH model is defined as follows:

$$r_t = \mu_t + \varepsilon_t$$

$$\varepsilon_t | F_{t-1} \sim N(0, H_t)$$

$$H_t = D_t R_t D_t$$

$$D_t = diag\left(\sqrt{h_{ii,t}}\right) i = 1, 2, \ldots n$$

where:

$r_t = (r_{1,t}, \ldots, r_{nt})$ is the vector of past observations of the return;

$\mu_t = (\mu_{1,t}, \ldots, \mu_{nt})$ is the vector of conditional returns;

$\varepsilon_t = (\varepsilon_{1,t}, \ldots, \varepsilon_{nt})$ is the vector of residual (assumed to be independent and identically distributed);

$F_{t-1}$ is the information set of the previous issue;

$H_t = \begin{bmatrix} h_{11,t} & \cdots & h_{1n,t} \\ \vdots & \ddots & \vdots \\ h_{n1,t} & \cdots & h_{nn,t} \end{bmatrix}$ is a conditional covariance matrix;

$R_t$ is a $N * N$ symmetric dynamic correlations matrix;

$D_t = diag\left(\sqrt{h_{ii,t}}\right) i = 1, 2, \ldots n$ is the four index's diagonal matrix of conditional standard deviations and each $h_{ii,t}$ calculated by *the model* $T - GARCH$ by which we can estimate the univariate process of each index.

To estimate the dynamic correlation coefficient, we decompose $R_t$:

$$R_t = Q_t^{*-1} Q_t Q_t^{*-1}$$

$$Q_t = (1 - \alpha - \beta)\overline{Q} + \alpha(\varepsilon_{t-1}\varepsilon_{t-1}^T) + \beta Q_{t-1}$$

$$\overline{Q} = T^{-1} \sum_{t=1}^{T} \varepsilon_t \varepsilon_t^T$$

where $Q_t^* = \begin{bmatrix} \sqrt{q_{11}} & \cdots & 0 \\ \vdots & \ddots & \vdots \\ 0 & \cdots & \sqrt{q_{nn}} \end{bmatrix}$ is a diagonal matrix of $Q_t$, $\overline{Q}$ is an unconditional variance matrix of standardized residuals, $z_t^T = D_t^{-1}\varepsilon_t$ is a vector of standardized residuals, $\alpha$ and $\beta$ is the coefficient of DCC and $\alpha + \beta < 1$. The larger the $\alpha$ and $\beta$, the more dependent the dynamic correlation coefficient of the current period on the previous period. It is worth noting that it is universally accepted that the return of the stock market is not symmetric [56]. If the leverage effect exists, the DCC-GARCH model cannot solve the phenomenon of asymmetric fluctuation of the second moment of stock return series in the empirical data of financial time series. In order to correct the asymmetry of volatility and its correlation under negative shocks, we further used the ADCC-GARCH model. ADCC-GARCH model introduces asymmetric terms based on Engle [7] model to consider the impact of negative information on correlation. On the one hand, it modifies the correlation coefficient of specific asset changes, and on the other hand, it describes the conditional

asymmetry of the correlation coefficient [57]. This allows us to more accurately observe the asymmetry of SP500, NASDAQ and USD under positive and negative shocks.

According to Sheppard et al. [57], we utilized the below model to incorporate asymmetry.

$$Q_t = \left(\overline{Q} - A^T\overline{Q}A - B^T\overline{Q}B - G^T\overline{R}G\right) + A^T\varepsilon_{t-1}\varepsilon_{t-1}^T A + G^T\delta_{t-1}\delta_{t-1}^T G + B^T Q_{t-1}B$$

$$\delta_t = I[\varepsilon_t < 0]\varepsilon_t$$

$$\overline{Q} = T^{-1}\sum_{t=1}^{T}\varepsilon_t\varepsilon_t^T$$

$$\overline{R} = T^{-1}\sum_{t=1}^{T}\delta_t\delta_t^T$$

where $A, B, G$ are $N * N$ coefficient matrix, representing the weight of each indicator in the current period, $I$ is a $N * 1$ indicator vector (if $\varepsilon_t < 0$, $I = 1$, otherwise $I = 0$)

It is worth noting that when $A, B, G$ are scalars, the model reduces to ADCC-GARCH:

$$Q_t = \left(\overline{Q} - \alpha^2\overline{Q} - \beta^2\overline{Q} - \gamma^2\overline{R}\right) + \alpha^2\varepsilon_{t-1}\varepsilon_{t-1}^T + \gamma^2\delta_{t-1}\delta_{t-1}^T + \beta^2 Q_{t-1}$$

$$\alpha^2 + \beta^2 + \lambda\gamma^2 < 1$$

where $\lambda$ is the maximum eigenvalue of matrix $\overline{Q}^{-1/2}\overline{R}\overline{Q}^{-1/2}$. According to the above constraints, we can estimate $\alpha$, $\beta$, $\gamma$.

In this study, a two-step approach is used to estimate the (A)DCC-GARCH models. At the first step, the GJR-GARCH model is estimated by using maximum likelihood estimation method for each asset. Then, we take the estimated values in the first step as given in the second stage thereby estimating the parameters in (A)DCC models [7]. The estimation of the ADCC-GARCH models were performed by using the R program and the *rmgarch* package in the R.

## 4. Data

Our data covers several indicative indexes including NASDAQ, USD index, SP500, and SHCI on a daily basis from January 2001 to July 2021, sourced from Yahoo Finance. The SP500 and SHCI are a common pair to be included in the US and China's stock market correlation studies due to the comprehensive coverage of the two indexes on the stocks in the two stock markets. However, with the increasing influence of technology on economies, NASDAQ become more impactful and, hence, influential to China's stock market [58]. We also included the USD index, the rate of currency in the international market with the lowest standard deviation, in our dataset as an indicator of a stable asset.

In order to investigate the co-movements between China's stock market and the US stock market, we constructed the daily return of the four indexes using following the formula:

$$r_t = \ln\frac{S_t}{S_{t-1}}$$

where $t$ represents a specific day from the sampling period from January 2001 to July 2021. $S_t$ denotes the daily closed price of a stock at day $t$.

We present the descriptive statistics of the four indexes in Table 1. The average return of USD return is close to zero with a minor standard deviation, reflecting USD to be a stable asset. The highest average return is NASDAQ, which is more than three times as much as SHCI, indicating the US tech stock market, to be more aggressive than the others. We distinguished that SHCI has the lowest return among the three stock markets but the highest standard deviation, which is against the investment theory that high risk corresponds with high return. This observation is consistent with Gao et al. [59], as investors in China are overly optimistic and there are speculators in China's stock market. Furthermore, through

the Jarque-Bera test, the four indexes are significantly different from a normal distribution. The three stock indexes have distributions with negative skewness and positive kurtosis, indicating fat tails and negative returns are more likely to happen.

**Table 1.** Data descriptive and statistics.

|  | SHCI | SP500 | NASDAQ | USD |
|---|---|---|---|---|
| Mean | 0.000103 | 0.000251 | 0.000372 | $-3.61 \times 10^{-5}$ |
| Median | 0.000550 | 0.000702 | 0.001037 | −0.000103 |
| Maximum | 0.094010 | 0.109572 | 0.132546 | 0.059653 |
| Minimum | −0.127636 | −0.137774 | −0.158689 | −0.027539 |
| Std.Dev. | 0.015808 | 0.012745 | 0.015306 | 0.005118 |
| Skewness | −0.407482 | −0.551040 | −0.327671 | 0.328351 |
| Kurtosis | 8.451031 | 16.22000 | 11.97649 | 8.641219 |
| Jarque-Bera | 6056.596 | 35086.63 | 16150.75 | 6430.763 |
| Probability | 0.000000 | 0.000000 | 0.000000 | 0.000000 |
| Sum | 0.493772 | 1.202683 | 1.781559 | −0.172838 |
| SumSq.Dev. | 1.195526 | 0.777088 | 1.120797 | 0.125299 |
| Observations | 4785 | 4785 | 4785 | 4785 |

The significant shocks in 2008 and 2020 shown in Figures 1 and 2 are caused by the global financial crisis (GFC) that occurred in 2008 and COVID-19 which took place abruptly in December 2019. The GFC leads to different degrees of economic contractions and recessions in all financial markets including the US, EU, Japan, and China [60]. However, China experienced lower volatility than the US in the financial market during the GFC due to the comparatively closed and bank-centered financial system [61]. During the pandemic, the US stock market underwent circuit breaks four times (March 9, March 12, March 16, and March 18). Nevertheless, the financial shocks lag with the actual side of the economy because of the information transmission. As Troster et al. [62] point out in their study, industry information has a predictive effect on the financial returns in the stock market.

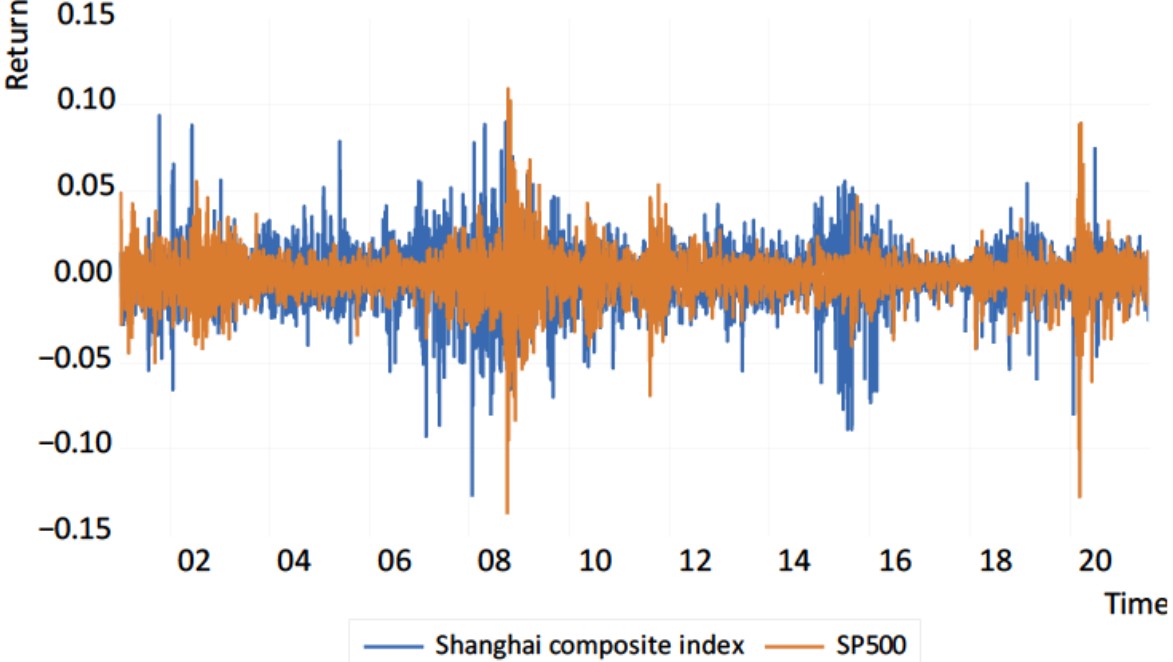

**Figure 1.** Log returns of SHCI and SP500 from January 2001 to July 2021.

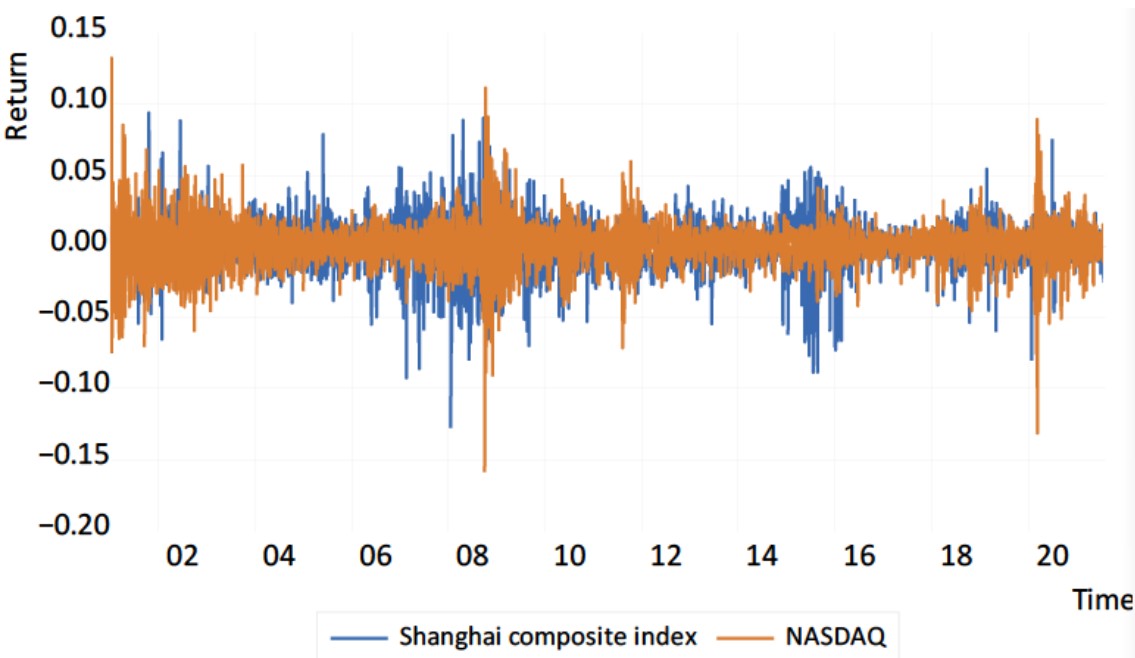

**Figure 2.** Log returns of SHCI and NASDAQ from January 2001 to July 2021.

The observations during the COVID-19 period show that the lags between SCHI and the US indexes exhibit a reverse relationship compared to the previous periods as the yield volatility in China matches the same day US indexes, indicating an increasing impact power of China during the pandemic and a mimic effect of the US stock market to China's one.

As is shown in Figure 3, USD shows a negative relationship with SHCI. This coincides with the previous study of Cao [63] in that the increase in the exchange rate of CHY is often considered a positive factor for China's stock market. Nevertheless, there are periods when SHCI deviates from the USD index. This could be a result from the stock crisis that took place in China in 2015.

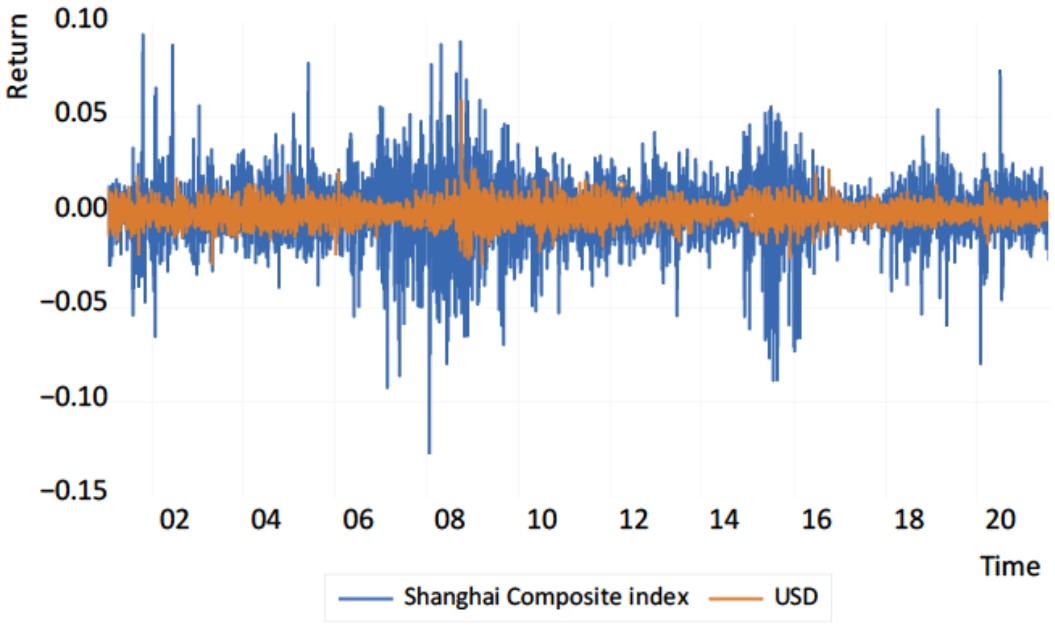

**Figure 3.** Log returns of SHCI and US dollar from January 2001 to July 2021.

## 5. Empirical Results

We used the bivariate (A)DCC-GARCH models to estimate the pairwise correlation between the Chinese and the US financial markets. Since this study only is interesting in the pairwise correlation, and the bivariate DCC models are easy to run in the R program, we repeatedly ran the bivariate (A)DCC-GARCH models in the R. The estimated results are presented in this Section. We first selected the best fit model for calculating the correlation of different pairs of the US market and China's market (Section 5.1). We then analyzed the parameters of the best fit model (Section 5.2). Section 5.3 discusses the correlation between the US market and China's market from an integral angle. In addition, we discuss the effect of different types of negative events on the correlation of the US and China's markets (Section 5.4).

### 5.1. The Information Criteria Values for the Models

Table 1 in Section 4 shows the characteristics of the returns of the four indexes. All four indexes are significantly different from the normal distribution. SP500, SHCI, and NASDAQ have negative skewness, while the USD has positive skewness. In order to find the best of the six models, we calculated the parameters of the ADCC GARCH and DCC GARCH with different distributions: multivariate normal distribution (MND), multivariate t distribution (MTD), and multivariate Laplace distribution (MLD). We used AIC, BIC, SBIC, and HQIC to test the effectiveness of the models. Tables 2–4 show the information criteria of the three index pairs. Tables 2–4 show the completeness of the information. Through the Tables, we can find the best fit model of the SHCI and SP500, and the pair of the SHCI and NASDAQ is the ADCC with MTD distribution. While the best match model of the SHCI and USD pair is DCC with MTD distribution.

**Table 2.** Information criteria of (A)DCC-GARCH models for SHCI and SP500 pair.

| Information Criteria | DCC-MND | ADCC-MND | DCC-MTD | ADCC-MTD | DCC-MLD | ADCC-MLD |
|---|---|---|---|---|---|---|
| Akaike | −12.210 | −12.209 | −12.340 | **−12.342** | −12.276 | −12.281 |
| Bayes | −12.192 | −12.190 | **−12.321** | −12.319 | −12.258 | −12.259 |
| Shibata | −12.210 | −12.209 | −12.340 | **−12.342** | −12.276 | −12.281 |
| Hannan-Quinn | −12.204 | −12.203 | −12.333 | **−12.334** | −12.269 | −12.273 |

Note: for each information criterion, the minimum value is in bold.

**Table 3.** Information criteria of (A)DCC-GARCH models for SHCI and NASDAQ pair.

| Information Criteria | DCC-MND | ADCC-MND | DCC-MTD | ADCC-MTD | DCC-MLD | ADCC-MLD |
|---|---|---|---|---|---|---|
| Akaike | −11.744 | −11.744 | −11.864 | **−11.867** | −11.789 | −11.795 |
| Bayes | −11.727 | −11.725 | **−11.845** | −11.844 | −11.772 | −11.773 |
| Shibata | −11.744 | −11.744 | −11.864 | **−11.867** | −11.789 | −11.795 |
| Hannan-Quinn | −11.738 | −11.737 | −11.857 | **−11.859** | −11.783 | −11.787 |

Note: for each information criterion, the minimum value is in bold.

**Table 4.** Information criteria of (A)DCC-GARCH models for SHCI and USD pair.

| Information Criteria | DCC-MND | ADCC-MND | DCC-MTD | ADCC- MTD | DCC-MLD | ADCC-MLD |
|---|---|---|---|---|---|---|
| Akaike | −13.585 | −12.209 | **−13.702** | −13.700 | −13.617 | −13.616 |
| Bayes | −13.567 | −12.190 | **−13.683** | −13.677 | −13.599 | −13.595 |
| Shibata | −13.585 | −12.209 | **−13.702** | −13.700 | −13.617 | −13.616 |
| Hannan-Quinn | −13.579 | −12.203 | **−13.695** | −13.692 | −13.611 | −13.609 |

Note: for each information criterion, the minimum value is in bold.

### 5.2. The Interpretation of the Estimated Results

Table 5 shows the parameters of the best fit models that we choose in Section 5.1. $\gamma_1$ of SHCI, SP500, and NASDAQ in Table 5 is extremely significant at the 5% level, indicating a leverage effect. $\alpha_2$ of the SP500 and NASDAQ is close to zero while $\alpha_2$ of SHCI is higher than 0.05 and significant, implying that investors in China are more contagious to the news than the investors in the US. It is worth noting that both values of g for t − 1 and t are not significant with large variance, demonstrating that the influence of bad news on the correlation is ambiguous. We will discuss this phenomenon in more detail in Section 5.4. Most of the parameters in Table 5 are significant, and the sum of the $\alpha_1$, $\alpha_2$, $\beta_1$ and $\frac{1}{2}\gamma_1$ of the three pair is all less than 1. Thus, the model is effective and stable.

**Table 5.** The estimated results of the best fit models.

| Explanatory Variables | ADCC-MVT SHCI + SP500 | DCC-MTD SHCI + USD | ADCC-MVT SHCI + NASDAQ |
|---|---|---|---|
| SHCI | | | |
| $\alpha_0$ | 0.000312 ** | 0.000113 | 0.000312 ** |
| | (0.000152) | (0.000175) | (0.000152) |
| $\alpha_1$ | 0.000001 | 0.000002 | 0.000001 |
| | (0.000001) | (0.000004) | (0.000001) |
| $\alpha_2$ | 0.054449 *** | 0.067948 *** | 0.054449 *** |
| | (0.008966) | (0.037635) | (0.008976) |
| $\beta_1$ | 0.933246 *** | 0.915254 *** | 0.933246 *** |
| | (0.008661) | (0.047597) | (0.008647) |
| $\gamma_1$ | 0.022333 ** | 0.022892 | 0.022333 ** |
| | (0.010738) | (0.025818) | (0.010732) |
| Shape1 | 4.400244 *** | | 4.400244 *** |
| | (0.310510) | | (0.311092) |
| SP500/USD/NASDAQ | SP500 | USD | NASDAQ |
| $\alpha_0$ | 0.000569 *** | −0.000054 | 0.000829 *** |
| | (0.000103) | (0.000064) | (0.000139) |
| $\alpha_1$ | 0.000002 | 0.000000 | 0.000003 |
| | (0.000001) | (0.000000) | (0.000002) |
| $\alpha_2$ | 0.000000 | 0.031241 *** | 0.000001 |
| | (0.011219) | (0.003448) | (0.007633) |
| $\beta_1$ | 0.884345 *** | 0.958642 *** | 0.895868 *** |
| | (0.023238) | (0.001626) | (0.018843) |
| $\gamma_1$ | 0.199700 *** | 0.014803* | 0.174673 *** |
| | (0.042032) | (0.008767) | (0.036266) |
| Shape2 | 5.756440 *** | | 6.500130 *** |
| | (0.518589) | | (0.592510) |
| A | 0.002712 | 0.003090 * | 0.001208 |
| | (0.008844) | (0.001660) | (0.001165) |
| B | 0.993701 *** | 0.991526 *** | 0.998768 *** |
| | (0.034701) | (0.004116) | (0.003243) |
| G | 0.000000 | | 0.000055 |
| | (0.002974) | | (0.000409) |
| a(t-1) | 0.013159 ** | 0.008734 | 0.011800 ** |
| B | 0.993701 *** | 0.991526 *** | 0.998768 *** |
| | (0.034701) | (0.004116) | (0.003243) |
| g(t-1) | 0.001151 | | 0.004301 |
| mshape | 5.770440 *** | 6.598618 *** | 6.012508 *** |
| | (0.299449) | (0.408857) | (0.292350) |

Note: t statistics in parentheses * $p < 0.1$, ** $p < 0.05$, *** $p < 0.01$.

### 5.3. General Correlations between China's and the US Financial Markets

Results of the best-fit models in Table 5 that measure the correlation between Chinese and the US financial markets suggest several strong dynamic relationships between the

two markets. These relationships are robust across the daily correlation of the SHCI to the NASDAQ, SP500, and USD, respectively, as shown in Figures 4–6. As illustrated in Figures 7–9, the results of the lag-one or intertemporal correlations on the one-to-one relationships between NASDAQ, SP500, USD at time $t − 1$, and SHCI at time $t$, also indicate strong relationships in the pairs. The lag-one correlation estimates are obtained by using (A)DCC models as well. Similar to the daily correlation, we simply need to estimate the (A)DCC models for the pairs of the SHCI at time $t$ and the US stock returns at time $t − 1$.

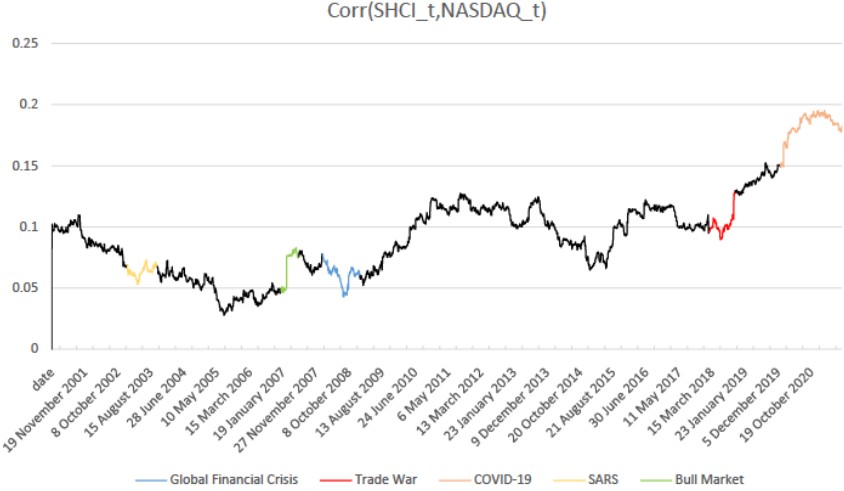

**Figure 4.** Dynamic correlation between SHCI and NASDAQ.

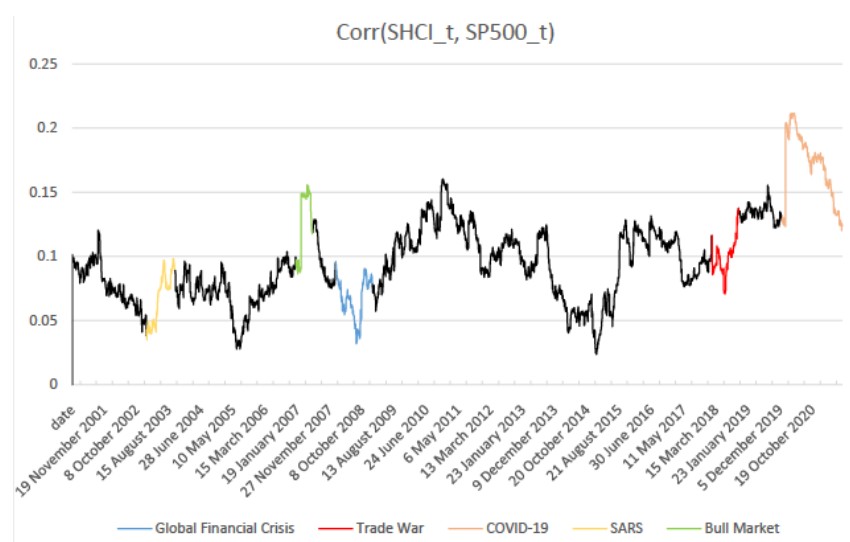

**Figure 5.** Dynamic correlation between SHCI and SP500.

We reported that the lag-one correlations between SHCI return on day $t$ and US stock market return on day $t − 1$, ranging from $−0.11$ to $0.30$, are stronger compared to the daily correlations between SHCI return on day $t$ and US's return on day $t$, ranging from 0 to 0.18. These results suggest that the US market movements are signals to Chinese investors, to some extent. Nevertheless, due to the time difference, the Shanghai exchange market (GMT +08:00) is about 12 or 13 h ahead of the US stock market (GMT−04:00). It is more apparent that the comparisons shown in Figures 4–6 indicate that SHCI has smaller impacts on the US stock market.

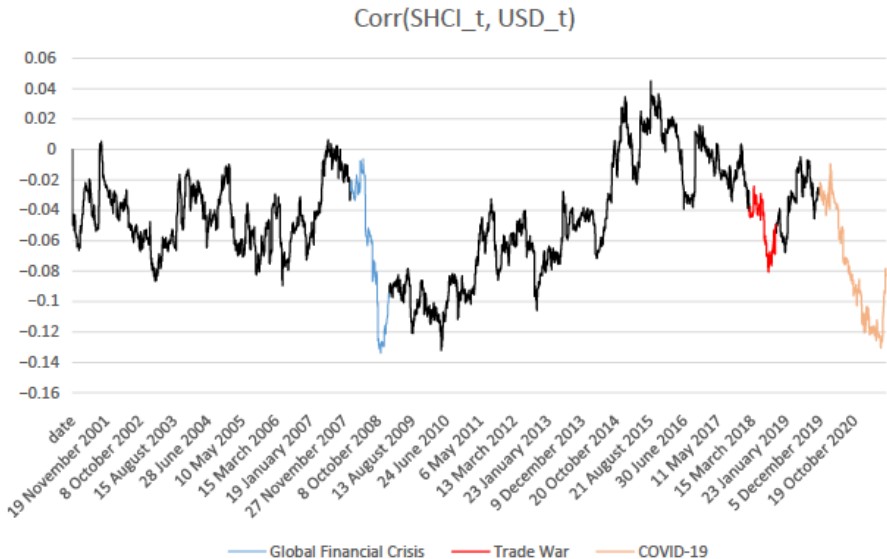

**Figure 6.** Dynamic correlation between SHCI and USD.

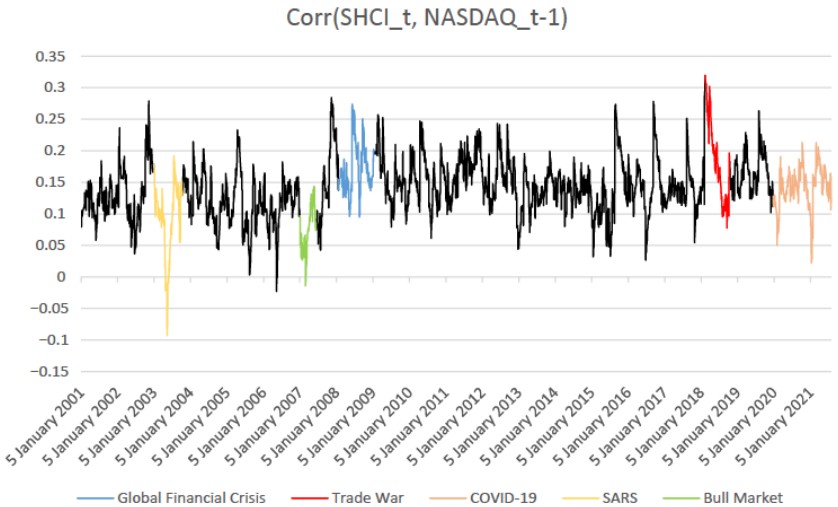

**Figure 7.** Dynamic correlation between SHCI at $t$ and NASDAQ at $t-1$.

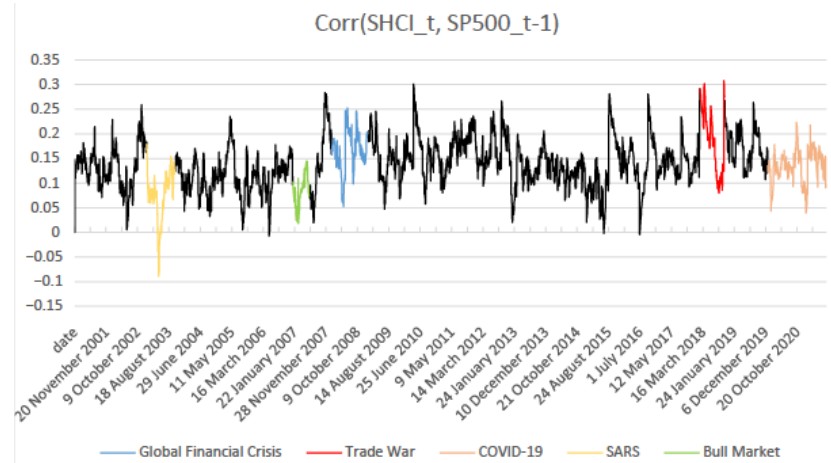

**Figure 8.** Dynamic correlation between SHCI at $t$ and SP500 at $t-1$.

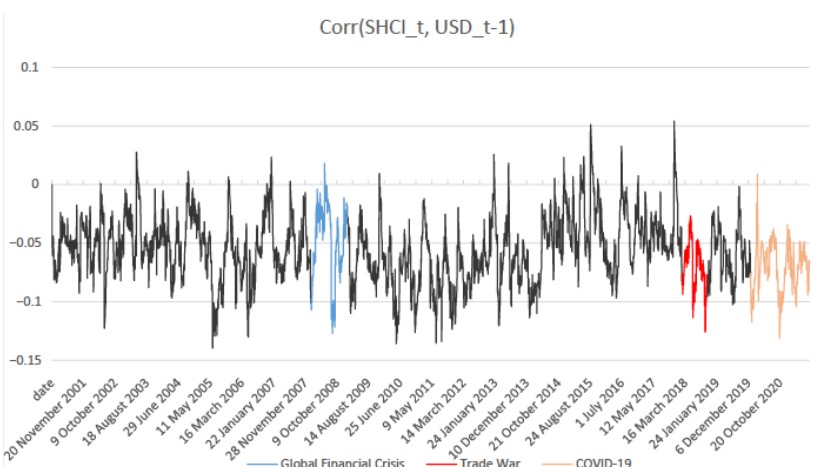

**Figure 9.** Dynamic correlation between SHCI at $t$ and USD at $t - 1$.

As shown in Figures 5 and 7, most of the lag-one correlation coefficients between the US stock market return, SP500, NASDAQ, and SHCI are larger than zero, indicating positive impacts of the returns on SP500 and NASDAQ on SHCI. Compared with the lag-one correlation, the daily correlation of the same stock pairs and the SHCI-USD pair present much weaker volatility but similar fluctuation trends over time. Several factors can result in the fluctuations including but not limited to global pandemics, global financial crises, or other major international events.

### 5.4. The Correlation Analysis of the Two Financial Markets

We analyzed the correlations between China's stock market and the US financial markets during five major events: severe acute respiratory syndrome (SARS) in China in December 2002; the volatile stock market in China around early 2007; the global financial crisis that took place from August 2007 to 2009; the US-China trade war that happened in 2018; and COVID-19, which outbroke in December 2019. Figures 4, 5, 7 and 8 show the NASDAQ and SP500 have similar trend during the occurrence of the four major events. Hence, our following analysis is largely based on SP500 and the ideology applies to NASDAQ as well.

#### 5.4.1. Analysis of Events before Financial Crisis

As shown in Figures 7 and 8, there are two abnormal points where the correlation of $t$ and $t - 1$ grows sharply, one in 2003, and the other in early 2007. China experienced a national epidemic caused by SARS in 2003. During the outbreak of SARS, it is clear that the lag-one correlation declined and further dropped to negative during the worst time, while the daily correlation increased. The correlation coefficients soon returned back to normal in June 2003 when the plague situation eased. This phenomenon indicates that China's stock market influences the US stock market under a national level pandemic situation where the reverse effect—the impact of the US stock market on China is negligible. We think this disconnect is caused by the intense fluctuation in China's stock market that reduces the assets in the financial market, and financial reallocation into lower-risk sectors [64], making investors less interested in the movements in foreign stock markets (a decrease in correlation between $SHCI_t$ and $US_{t-1}$). In addition, as the investors in the US are more concerned about the epidemic and its impact on the domestic financial market and hence are seeing the China's stock market performance as a strong signal, leading to the daily correlation ($SCHI_t$ and $US_t$) greater. The daily correlation between China and the US recuperates from the sharp increase when the national major event emergency level in China is lower.

A similar situation is observed during the bull market in China in early 2007. The SHCI broke the 3000 points for the first time on 26 February 2007. Spurred by the positive

news, the investors in China increased their capital investment in the stock market, and China's stock market started to escalate faster, when the lag-one correlation ($SCHI_t$ and $US_{t-1}$) between the US stock return and China's stock return dropped to below 0.05 and maintained under 0.1 level while the daily correlation ($SCHI_t$ and $US_t$) stayed above 0.14 for approximately two months. This is consistent with the study of Gao et al. [59] as investors in China are usually optimistic, and there are speculators in China's stock market to increase investment in the flourishing market, which further pushes the market to even higher levels. However, this situation soon ended in May 2007 as the Chinese government realized the bubble boom and increased the stamp duty from 1‰ to 3‰. This increase was seen as an implementation to curb speculations in the exchange market [65]. The lag-one correlation ($SCHI_t$ and $US_{t-1}$) increased temporarily as investors in China became more relational on 30 May 2007. The volatile correlation shows that investors in China are more contagious to the news as mentioned in session 5.2 and do not evaluate the stock price from a long-run vision compared to investors in the US. Hence, it is clear that national shocks have a short-term effect on correlations between China and the US stock market and the directions of impacts on daily and the lag-one correlation are opposite.

5.4.2. Analysis of Events after Financial Crisis

Our results show that while the SARS is likely to cause opposite direction movements of daily and the lag-one correlation between China and the US stock markets, global events, such as the 2008 financial crisis and the COVID-19 pandemic, may tend to have a different impact on daily and the lag-one correlations.

As shown in Figures 5 and 8, there are considerable decreases in both the daily and the lag-one correlation in early 2008. The global financial crisis caused by the collapse of subprime housing mortgages is accepted as the cause of higher correlations [66]. The reasons for the latter decrease are not conclusive in the literature, but we posit that the fundamental difference between the two systems of the housing markets is the main cause; the US market is more market-oriented compared to China. The vigorous policy support gives the public confidence in the housing market in China and, hence, China's housing market is unlikely to experience a subprime mortgage crisis as the US did [67]. The surging increase in property investments also gives the homebuyers confidence in the housing price (Figure 10).

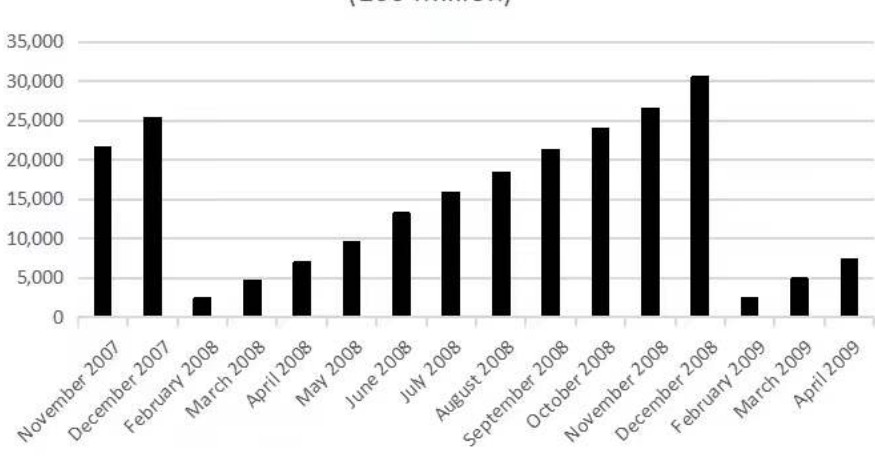

**Figure 10.** Real estate development investment in China. Data source: National Bureau of Statistics [68].

The correlation between the pair of USD and SHCI have increased. However, the abnormally low correlation between China and the US stock markets did not last for long. The US market regulators and operators issued a series of policies to stabilize the market

and prevent further damage from the crisis. The Federal Housing Finance Agency (FHFA) took over Fannie Mae and Freddie Mac Meanwhile on 6 September 2008. Meanwhile, the Federal bank announced the federal fund rate to be in the range of 0 to 0.25% in December 2008. The correlation was seen to stabilize and converge to its long-term level.

Based on the comparison between the correlation coefficient before and after the crisis, it is clear that efficient information transmission between exchange markets requires a stable international environment. This condition requirement can also be found during the US-China trade war and the COVID-19.

During the US-China trade war, the correlation coefficients of the two stock markets dropped ominously, starting from the news announcement of the releasing report on imposing of importing tariffs on Chinese goods by Commerce Secretary Wilbur Ross. The lag-one correlation decreased more drastically with the escalation of the conflict. Results from the literature are conclusive that the trade war was the cause of the dropping correlation between the two stock markets [69]. The correlation returned back to its normal level with signs of alleviation of the US-China tension after the continuous negotiations, and the release of new announcements starting from August 2018.

The effect of COVID-19 on the correlation coefficients between the two markets is a bit ambiguous compared to the clear, unified impacts of the GFC and trade war. COVID-19 was a bit strange. The daily correlation increased, while the lag-one correlation decreased in the early stage which somehow indicated that the US stock market is on the information accepting side, similar to the period of SARS and the bull market in 2007, as mentioned in Section 5.4.1. However, both types of correlation coefficients decreased in mid-2020. We believe this unintuitive pattern of correlations during COVID is caused by the quick, worldwide spread of COVID-19. The virus was first considered only a national major event and, hence, the effect was similar to SARS and the bull market in China in 2007. However, it soon grew into a global pandemic that followed the same pattern (international major event) as the 2008 global financial crisis and the US-China trade war.

Figures 6 and 9 show that highly negative correlations between the USD index and SHCI can be observed during global events. This effect can be seen in both the daily correlation and lag-one correlation. We think this phenomenon is caused by the safe-haven nature of the USD, as investors prefer to hold the US dollar during crises and are less willing to invest in volatile markets, such as stock markets based on Keynesian theory [70].

## 6. Conclusions and Discussion

### 6.1. Conclusions

This study measured the dynamic correlation between the returns on the US financial market and China's stock market and discussed the trend changes of their correlation during several Black Swan Events. The ADCC-GARCH (1,1) and DCC-GARCH (1,1) models were employed to evaluate the time-varying daily and the lag-one correlation in three index pairs: (1) SHCI and SP500; (2) SHCI and NASDAQ; (3) SHCI and USD index. Most hypotheses have been empirically supported in this study. We found that the correlations between the Chinese and US financial markets are time-varying. During the COVID-19 pandemic, their daily correlations were strengthened, but it was the opposite during the financial crisis.

Empirical results of the past negative events show that the co-movement between China's stock market and the US stock market is varied, depending on the type of the major event. The daily correlation and lag-one correlation usually move oppositely when there is a national event, indicating that the information transmitted to the stock market is a one-way propagation. On the other hand, the effect of a negative global event decreases both the daily and the lag-one correlation where the correlation might decline to a low level, meaning the co-movement of the two stock markets will shrink and the two markets behave like two independent markets.

*6.2. Discussion*

On the basis of the dynamic correlation between Chinese and the US stock markets and the Chinese stock market and the U.S. dollar, we can further infer new findings and discussions. We found that the correlation between the Chinese and the US stock markets varies during different black swan events. For example, in the early stages of the COVID-19 pandemic, Chinese stock markets were hit first, and then the US. The correlation between the Chinese and U.S. stock markets has grown markedly during the pandemic, suggesting that the Chinese stock market has become more influential to the U.S. stock market. The change in the lag-one correlation indicates that the impact of the US stock market on the Chinese stock market does not change much as a whole. We may think that the pandemic has not amplified the impact of the US on the Chinese stock market, however, during the COVID-19 pandemic and the financial crisis, the US dollar index has a significantly larger negative correlation to the returns of the Chinese stock market, which confirms that the appreciation of the US dollar can lead to the flow of funds from the Chinese stock market to the US dollar. Chinese and U.S. stock markets have become more interdependent, or integrated, during the pandemic. In this way, China and the US should consider each other's risk shocks in terms of financial risk prevention. Although the appreciation of the US dollar can play a role in reducing US inflation, we also have to consider the impact on China. Once the shock is too large, it may backfire on the US stock market.

The findings of this study are conducive to helping investors judge the impact of negative events in order to make a better investment strategy, and help governments publish corresponding policies to mitigate the negative effect of the events. We suggest that investors be mindful of making transnational hedges (investing in stock markets) but consider investing in currency indices to hedge the risk of great negative events given that the co-movement of the two stock markets decreases, but the negative correlation of USD and SHCI is usually enhanced during global negative events. This paper also delivers valuable insights for investors to evaluate the length of a crisis, where the duration of the impact on the correlation between the two markets is usually shorter than a global major event. On the policy side, we recommend that Chinese regulators and operators should pay more attention to the externality of the policy compared to the US, given the higher sensitivity of China's stock market.

The analysis of this paper also has some limitations. Firstly, the correlation analysis at different stages is qualitative. We did not use quantitative methods to reflect the impact of the COVID-19 epidemic and the financial crisis on the correlation between Chinese and American stock markets. Second, the nonlinear correlation between financial markets has also been widely concerned [71,72]. This paper only measures the linear correlation between the Chinese stock market and the US financial markets. In the future research, the nonlinear correlation between them is worth exploring. In addition, it is also worthwhile to quantify the impact of the Black Swan Events on the co-movement between the Chinese and American financial markets.

**Author Contributions:** Conceptualization, J.L. and Y.W.; methodology, J.L. and Y.W.; software, J.L.; validation, S.S. and S.Q.; resources, J.L.; data curation, R.Q. and S.S.; writing—original draft preparation, Y.W.; writing—review and editing, S.Q. and J.L.; visualization, R.Q.; supervision, S.S.; project administration, J.L.; funding acquisition, S.S. All authors have read and agreed to the published version of the manuscript.

**Funding:** This research received no external funding.

**Institutional Review Board Statement:** Not applicable.

**Informed Consent Statement:** Not applicable.

**Data Availability Statement:** All data can be obtained by email from the corresponding author.

**Acknowledgments:** This work has been assisted by the China–ASEAN High-Quality Development Research Center at Shandong University of Finance and Economics and the "Theoretical Economics Research Innovation Team" of the Youth Innovation Talent Introduction and Education Plan of

Colleges and Universities in Shandong Province for financial support, as well as the Faculty of Economics and the Centre of Excellence in Econometrics at Chiang Mai University.

**Conflicts of Interest:** The authors declare no conflict of interest.

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
