# Peer review of "Dynamic Correlation between the Chinese and the US Financial Markets: From Global Financial Crisis to COVID-19 Pandemic"

_axioms, doi:10.3390/axioms12010014_

Round 1

Reviewer 1 Report

The study is interesting, but some improvements are necessary.

The introduction does not clearly state the objective of the research. There are two research questions, but it is not clear why these are interesting to the reader. The authors should clearly explain not only the gap in knowledge, but also the contribution of the present study (in the introduction and the conclusions).

There is no hypothesis for this study. What do the authors expect to find, in the light of the previous literature? How do they plan to explain the reasons for the differences between the two markets? Has this question been asked before? The literature is very thin and not compelling.

The methodology and results clearly need some hypotheses. Right now, the analysis is not anchored to anything. In my opinion, it would be more logical to have the Methodology before the Data. Because, after data presentation, we expect the results, not more formulas. The section "5.1 Model selection" appears to belong to the methodology, not the results -- or the title of this section is not appropriate.

Which are Model 3 and Model 4 from Table 5? The models are not numbered before that. The title of section 5.2 should tell something relevant, not refer to a statistical method.

Only the national events in China are analyzed (5.4.1.). What about the national events in the US? We are talking about a correlation, so we should see both sides of the issue.

Coming back to the beginning of the review, we need to refer to some hypothesis. This is quantitative research so just mentioning the research question is not enough. Clear hypotheses are needed and referred to again at the end of the discussion. The presentation of national and international events is interesting, but the discussion is not very structured and seems anecdotal. Some clear references and a much better structure for the discussion sections is needed. A summary table of causes and effects would improve the discussion.

The research has a clear bias in presenting the results. I do not think this is good practice, considering that the title indicates a "dynamic correlation" between A and B. But we talk mostly about A. What about B? The results and discussion should be presented in a more balanced manner. Right now, it looks biased.

Reviewer 2 Report

The manuscript deals with the dynamic correlation between Chinese and US financial markets using a modified Asymmetric Dynamic Conditional Correlation Generalized Autoregressive Conditional Heteroscedasticity (ADCC-GARCH model). The authors investigated this model from the aspect of different multivariate distributions, and its application was made by compiling the Shanghai Composite Index (SHCI) with different indices of the US financial markets (SP500, NASDAQ and the US dollar index). The use of GARCH-based models in solving such econometric problems has been fully justified for decades. Therefore, from the aspect of solving the aforementioned correlation problem, as well as the results obtained in this way, the manuscript seems interesting, although it does not contain too many mathematical elements. Therefore, I believe that it can be taken into account in this form, with certain corrections by the authors, from which I single out the following:

1. The authors should be more specific and indicate still in the introductory part the motivation for using the ADCC-GRACH model, as well as its advantage over the symmetric DCC-GARCH.

2. Pg. 4: It is natural to use the more precise, "mathematical" notation for the log-returns ht, which authors use later in Eq. (1).

3. Pg. 7: Please, explain the abbreviation "GJR". On the same page, in the phrase "εt is normally distributed or student-t" should be more precisely describe that it is a series of innovations (or "white noise").

4. Pg. 8: The sentence "The leverage effect exists" is not the clearest. Please "connect" it to the previous sentence.

5. Pg. 10: Please, explain the abbreviations for the Akaike, Bayes, Shibata, and Hannan-Kuinn information criterion(s).

6. On the same page, Table 4: The optimal Bayesian information criterion is -11.845 (not -11.844).

Reviewer 3 Report

This study presents the results of an empirical analysis of the differential response of the Chinese stock market and the US stock market to the COVID-19 pandemic.

Regarding the results of this analysis, the authors state that the reason is that the two markets are independent of each other. However, existing literature has suggested a phenomenon of co-movement in capital markets around the world. However, this study does not. On the other hand, the COVID-19 pandemic is a global event, and although the reaction of the capital market due to this may have a time-lagging effect, it is expected that there will be market synchronism.

Therefore, this study needs a sufficient, logical and systematic explanation as the analysis results are different from those of the existing literature.

Reviewer 4 Report

This paper provides an empirical analysis on the relationship between Chinese and US financial markets. The Chinese financial market is analyzed in terms of Shangai Index, whereas for US market SP500, NASDAQ and Dollar indices are considered. The results could be interesting from investors perspective and the methodology used for studying conditional correlation is somehow correct. I have few remarks.

- First of all, it is not clear why only stock market is considered for China and three alternative quantities are instead considered for US. The main problem I have in understanding this choice is the inclusion of exchange rate index. The relationship between Chinese stock market and US exchange rate should be exploited with more detail;

- The way which methodology (Section 4) is written is a bit ambiguous. Do the authors use a classical 2-step approach for DCC estimation? Why they state that first volatility is estimated and, then, the correlations?

- The authors should explain why they estimated separated bivariate DCC models instead of a single multivariate model;

- It is not clear how these figures 5, 7 and 9 are obtained. Can the authors be more specific?

- I would ask the authors to specify that all the claims in Section 5.4 are given by qualitative inspection of the results. Indeed no "even study" is conducted here to assess the impact of events on correlations.

- I don't see how the following sentences in the abstract "the U.S. and China’s stock markets have no overlap in their trading hours, this study measures all possible correlations between the current period and one period lag" are related. Consider removing this sentence in the abstract. Please, explain better in the text this aspect of the paper that is unclear.

- In the title consider writing just one time the word "financial market", e.g.  "Chinese and the US financial markets"

Round 2

Reviewer 1 Report

The paper has significantly improved. Congratulations!

I would suggest to the authors some refinements to the Literature review and the discussion overall. In the literature review, previous contributions are presented for several types of markets, not just for the stock market. These should be separated. That is: contributions regarding the stock market grouped together, contributions regarding currency and commodity markets separately. In the present version, all these markets are in one big pile of literature, which is not ok. Also, the authors could introduce some subsections for the literature review. 

One important thing that is missing: The hypothesis (or plural, if it is necessary). I recommend the authors to add a hypothesis and to refer back to this hypothesis in the Discussion.

Good luck with this revision!

Reviewer 3 Report

This version of the manuscript reflects the results of the first review well. However, the presentation needs to be rearranged from the reader's point of view.

Author Response

Dear Professor,

Thank you for giving us the opportunity to submit a revised copy of the manuscript. We appreciate the time and effort that you have dedicated to providing your valuable feedback on this manuscript again. Your comments and concerns are highly insightful and enabled us to improve the quality of the manuscript. 

According to your suggestions and comments, we have made some changes in the literature review and the discussion. In order to facilitate readers' reading and understanding, we divided the literature review into three parts, and added some descriptions corresponding to the research hypotheses. In the discussion section, we also made some cohesive modifications.

We hope that this revised version has satisfactorily addressed all of your concerns. 

Thanks again.